# Charge Transport Mechanisms of Black Diamond at Cryogenic Temperatures

**DOI:** 10.3390/nano12132253

**Published:** 2022-06-30

**Authors:** Andrea Orsini, Daniele Barettin, Federica Ercoli, Maria Cristina Rossi, Sara Pettinato, Stefano Salvatori, Alessio Mezzi, Riccardo Polini, Alessandro Bellucci, Matteo Mastellone, Marco Girolami, Veronica Valentini, Stefano Orlando, Daniele Maria Trucchi

**Affiliations:** 1Università degli Studi Niccolò Cusano, 00166 Roma, Italy; daniele.barettin@unicusano.it (D.B.); sara.pettinato@unicusano.it (S.P.); stefano.salvatori@unicusano.it (S.S.); 2Università degli Studi Roma Tre, 00154 Roma, Italy; fma.ercoli@gmail.com (F.E.); mcrossi@uniroma3.it (M.C.R.); 3Istituto di Struttura della Materia, ISM-CNR, 00015 Monterotondo Stazione, Italy; polini@uniroma2.it (R.P.); alessandro.bellucci@ism.cnr.it (A.B.); matteo.mastellone@ism.cnr.it (M.M.); marco.girolami@ism.cnr.it (M.G.); veronica.valentini@ism.cnr.it (V.V.); stefano.orlando@ism.cnr.it (S.O.); daniele.trucchi@ism.cnr.it (D.M.T.); 4Istituto per lo Studio di Materiali Nanostrutturati, ISMN-CNR, 00015 Monterotondo Stazione, Italy; alessio.mezzi@ismn.cnr.it; 5Università degli Studi di Roma “Tor Vergata”, 00133 Roma, Italy

**Keywords:** black diamond, LIPSS, cryogenic temperatures, electric conductivity, activation energy, variable range hopping

## Abstract

Black diamond is an emerging material for solar applications. The femtosecond laser surface treatment of pristine transparent diamond allows the solar absorptance to be increased to values greater than 90% from semi-transparency conditions. In addition, the defects introduced by fs-laser treatment strongly increase the diamond surface electrical conductivity and a very-low activation energy is observed at room temperature. In this work, the investigation of electronic transport mechanisms of a fs-laser nanotextured diamond surface is reported. The charge transport was studied down to cryogenic temperatures, in the 30–300 K range. The samples show an activation energy of a few tens of meV in the highest temperature interval and for T < 50 K, the activation energy diminishes to a few meV. Moreover, thanks to fast cycles of measurement, we noticed that the black-diamond samples also seem to show a behavior close to ferromagnetic materials, suggesting electron spin influence over the transport properties. The mentioned properties open a new perspective in designing novel diamond-based biosensors and a deep knowledge of the charge-carrier transport in black diamond becomes fundamental.

## 1. Introduction

Diamond is characterized by excellent physical and chemical properties. It has the highest thermal conductivity, as well as the highest mechanical hardness, among the solids. It is a wide bandgap semiconductor (5.47 eV) with a high mobility for both electrons and holes and it displays a very high radiation hardness. Boron-doped diamond shows superconductivity properties, as demonstrated by high-temperature, high-pressure (HPHT) synthetic samples [1]. Applications of diamond range from high-power electronics to particle and radiation detection [2,3,4,5,6,7,8,9], as well as solar energy conversion, thanks to the effect of thermionic emission at relatively high temperatures [10], that can overcome the efficiency limits dictated by the Shockley–Queisser law [11]. It was indeed demonstrated that diamond-based thermionic emitters can operate at temperatures as low as 520 K [12]. Diamond also received an increased attention as electrochemical transduction material for the fabrication of biosensors and lab-on-chip devices [13]. The chemical vapor deposition (CVD) represents the mature technique employed for the growth of ultra-high-quality synthetic diamond. In addition, the fs-laser treatment of diamond samples is rapidly growing as an easily applicable technique to finely define the morphology of the material surface by providing a way to fabricate nano-patterns, even at a sub-wavelength periodicity [14].

In the last few decades, surface treatments using femtosecond lasers showed significant technological progress that enabled the development of effective techniques for nanostructuring the surface of both single-crystal and polycrystalline diamond. Indeed, by irradiating the diamond surface with ultra-short pulses at a precise energy density, it is possible to induce periodic patterns also known as LIPSS (Laser-Induced Periodic Surface Structures) with a periodicity that can be controlled by the laser wavelength [15,16]. The modification of the diamond surface is not solely morphological, but as several studies have recently demonstrated, it also improves the optical absorption property of the material [17]. Femtosecond laser-treated diamond, named “Black Diamond” (BD), is able to combine a very high optical absorptance with a very high mobility-lifetime product of the charge carriers, i.e., electrons and holes [18], thus launching the nanotextured black diamond as a possible absorbing layer for high-efficiency solar cells and for the fabrication of selective solar receivers. It is important to point out that femtosecond laser irradiation with an intensity high enough to locally induce avalanche photoionization, which consequently causes an irreparable optical breakdown within the diamond bulk, is able to also induce diamond graphitization. Therefore, by using fs-laser processing, it is possible to create conductive graphite paths embedded into the diamond bulk, opening the way for device fabrication with three-dimensional contacts [19,20]. However, even if heating is excluded due to the extremely short pulse duration, such fs-laser processing may induce stress and create defects in the diamond lattice near the surface as well as into the bulk [21]. In this paper, we regulated the laser parameters (i.e., fluence, repetition rate, and processing time) to induce periodic surface structures (LIPSS) (see Figure 1) and we split the treatment into two different orthogonal steps to better control the necessary quantity of defects within the diamond bandgap.

Since the diamond is hygroscopic and the BD surface area is far larger than the geometric area of the pristine diamond plates, it is reasonable that airborne molecules can significantly be adsorbed on the black diamond surface. In fact, once the process of the BD fabrication is terminated, an increase of the surface conductivity with the laser treatment proceeds is observed through two main steps:(i)outgassing of molecules absorbed at the surface in a relatively short time (a few hours);(ii)stress relaxation of the nanostructured BD lattice, which takes up to several days as revealed by Raman spectroscopy measurements.

After a temperature stimulation up to 800 K, the lowest sheet resistance (25 kΩ/sq.) at room temperature (RT) was found for a sample tagged 2T-8020 (see Section 2 for tag explanation, Appendix A), due to lattice relaxation verified by Raman diamonds’ peak shifts around 1332 cm^−1^ [23]. Samples with a higher percentage of accumulated fluence in the second treatment are more insulated and samples with the laser treatment in only one direction are much more insulated.

The analysis on the electronic properties of nanostructured BD surfaces is of particular relevance, especially if extended at cryogenic temperatures, where the transport mechanisms due to defects can be disentangled from the thermally driven ones. This work aims at illustrating the charge transport mechanisms that determine the conduction in diamond samples with dual-textured nanostructured surfaces unveiling their peculiar charge transport properties.

## 2. Materials and Methods

The samples used in this work were freestanding 10 × 10 × 0.25 mm^3^ thermal-grade polycrystalline diamond plates (Element Six Ltd., Didcot, UK). The surface texturing was performed by means of an ultra-short pulse laser. The femtosecond-pulse (100 fs) laser beam was focused perpendicularly to the CVD-diamond surface in a high-vacuum chamber (10^−7^ mbar). The laser wavelength was 800 nm and the pulse energy was kept constant at 3.6 mJ for a power density higher than the diamond ablation threshold (2 TW/cm^2^). The spot size was 250 μm in diameter, inducing a similar modification region on the sample and an ablation region of about 150 μm. Double-nanotexuring of the diamond surface along the X-Y directions was obtained following the two-step boustrophedic procedure described in [22]. A complete scan of the sample surface was performed by means of a computer-controlled micrometric X-Y translational stage. In the first step, a one-dimensional LIPSS structure was defined along the X direction. In the second step instead, the LIPSS structure was defined along the Y direction. After the process, a spatially crossed LIPSS resulted from the superposition of the two-step procedure. The scan speed was fixed to obtain the best light-absorptance in the visible range [17]. In such a case, the accumulated fluence resulted to be 12.5 kJ/cm^2^. In this work, two BD samples have been investigated. The first sample, named 2T-9010, received 90% of the total energy during the first scan along the X direction and the remaining 10% during the last scan along the Y direction. For the second sample, named 2T-8020, the percentages of energy released during the X and Y scans were 80% and 20%, respectively. After the laser treatment, both samples were immersed in a strongly oxidizing solution (H_2_SO_4_:HClO_4_:HNO_3_ in the 1:1:1 ratio) for 15 min at the boiling point in order to remove any debris, as well as non-diamond carbon phases. Finally, the samples were cleaned by ultrasound sonication and rinsed in deionized water. Surface morphology was studied by a Zeiss LEO Supra 35 field emission gun scanning electron microscope (FEG SEM). Secondary electron images were acquired with an in-lens detector. The electron beam energy was set at 10 keV.

Raman spectroscopy was performed in back-scattering geometry by using a Dilor XY triple spectrometer (1 cm^−1^ resolution), equipped with an Ar^+^ laser (514.5 nm), cooled CCD detector, and an adapted Olympus microscope arranged in confocal mode. The spot size was 2 mm and 20 intensity points were collected to provide a suitable statistic. Line widths and intensities were determined using the Thermo-Grams Suite v9.2 software. 

The surface chemical composition of the sample was investigated by X-ray photoelectron spectroscopy (XPS). The XPS investigations were performed by an Escalab 250Xi spectrometer (Thermo Fisher Scientific, Altrincham, UK), equipped with a monochromatized Al Kα source and 6-channeltron detection system for spectroscopy. A selected area of 1 mm in diameter was analysed. The spectral regions with high resolution were registered operating at a constant pass energy of 40 eV, while the survey scan and the C KLL Auger spectrum were collected at 100 eV pass energy. All XPS data were collected and processed by the Avantage v.5 software (Thermo Fisher Scientific, Altrincham, UK). More details are reported elsewhere [24].

A Kapton sheet, 50 µm thick, was used as shadow-mask for thermal evaporation of the silver contacts on each sample. Four holes of 1 mm diameter were drilled on the sheet so that the silver dots were located at the vertices of the squared sample. Electronic transport measurements were performed at cryogenic temperatures by using a helium closed-cycle cryostat (Galileo, Italy) equipped with an Oxford ITC4 temperature controller and pumping system at 10^−6^ Torr vacuum level. Both samples were fixed to a thin copper plate (0.5 mm thick) with silver paste in order to assure a good thermal contact with the cryostat cold finger. An ultra-thin mica foil was used as an interlayer, guaranteeing electrical insulation during the measurements. 

The resistivity of the samples was measured following the Van der Pauw method. Therefore, for each sample, four contacts were created at the edges of the conductive surface and numbered from 1 to 4 according to a counter-clockwise rotation aligning the LIPSS nano-waves direction to the 1 → 2 and 3 → 4 sides. A Keithley 2700 source/m, equipped with a Keithley 7700 20 channel multiplexing board, was used to change the connection between the ohmmeter and the sample contacts. The multiplexing board was programmed to inject a current within two contacts on one side and to read the voltage between the two remaining contacts located on the opposite side of the sample. Once the desired temperature was reached, a four-phase cycle was employed, so the injection (reading) scans were performed following a counter-clockwise order: (i) 1–2 (3–4); (ii) 2–3 (4–1); (iii) 3–4 (1–2); and (iv) 4–1 (2–3). 

## 3. Results

### 3.1. FEG SEM Characterization

A high-resolution FEG SEM was used to analyze the effect induced by the fs-laser on the sample surface. As stated in the introduction, the fs-pulsed laser is able to create LIPSS. The linearly polarized beam induces a one-dimensional LIPSS during a scan. A nanotexturing “ripple wave” is perpendicularly oriented to the laser-light polarization. Due to the interaction of the laser-pulse with the induced plasma at the surface, LIPSS nanotexturing periodicity (λ_LIPSS_) depends on the laser wavelength λ_LASER_ and on the refractive index of diamond n_d_. It can be estimated by [15]:λ_LIPSS_ = λ_LASER_/(2 n_d_)(1)

During the scan along the X direction, the refractive index of the sample is equal to the value of pristine diamond (n_d_ = 2.4). In this case, for λ_LASER_ = 800 nm, a surface periodicity of about 170 nm is estimated. During the scan along the Y direction, the nanotexturing structure is still modified according to the energy released by the laser beam. However, in this case, an increase of the absorptance also at 800 nm must be considered. It is worth noting that the percentage of energy released during the second Y-scan is not equal to 1/9 (for sample 2T-9010) and 1/4 (for sample 2T-8020) of the energy released by the X-scan. In particular, it was evaluated that for the 2T-8020, the two scans released almost the same amount of energy due to the increase of the absorptance induced by treatment itself (incubation effect) [22]. As demonstrated in [25], the roughness of the laser-treated samples is around 500 nm and the 2T-9010 mainly resembles the structures of a one-dimensional treated-sample (1T) under the same laser system, while the 2T-8020 shows a major degradation of the LIPSS created in the first step of the fs-laser process (see Figure 2).

At the lower magnification (Figure 2a,c), a periodicity of about 170 nm is clearly revealed. For both the 2T-9010 and 2T-8020 samples, the energy released by the laser during the Y scan is able to modify the periodicity induced during the X-scan (the 10% and the 20% of the total energy, respectively). In the 2T-8020 sample, the reduction of the nanostructuring alignment is more evident in comparison to what is observed for the 2T-9010. The reduction of the periodicity formed by the first X-scan unavoidably leads to a slight broadening and a loss of continuity in the ripples and a major presence of diagonally oriented interrupting grooves (see orange lines of Figure 2c). Moreover, the distance between the interrupting grooves is much larger than the 170 nm width of the ripples. Although a deep understanding of the phenomena underlying this effect requires further investigations, we infer that said effect could be attributed to a strong reduction of the refractive index of the diamond-surface after the first X-scan.

### 3.2. Surface Chemical Characterization

After the fs-laser treatment, the diamond surface was covered by an amorphous layer, as reported in the literature [26]. Therefore, we immersed the samples in a strongly oxidizing solution to etch the non-diamond domains. In order to perform a comprehensive surface chemistry bonding structure characterization, aimed at investigating the laser-induced structural modifications and the possible residual presence of amorphous carbon and graphite clusters, we analyzed BD samples by XPS and Raman techniques.

The XPS survey scan shown in Figure 3a demonstrates the presence of C and a small amount of O (~2 at.%), assigned to surface contamination. The peak-fitting analysis of C 1s signal was performed by introducing two synthetic peaks positioned at BE = 284.2 eV and 285.2 eV, assigned to the C–C bond and aliphatic carbon, respectively. As it is reported in reference [27], the peak-fitting analysis of the C 1s signal for carbon-based materials can lead to ambiguous conclusions, because the range of value for C–C hybridized sp^2^ and C–C hybridized sp^3^, is not well established. However, a more accurate method used to determine the carbon hybridization (sp^3^/sp^2^) state in carbon-based materials is based on the calculation of the D parameter. This parameter is defined as the distance between the most positive maximum and the most negative minimum of the first-derivative C KLL spectrum, and it is linearly dependent from the amount of C sp^2^ or C sp^3^ [23]. In our case, the obtained D value was 13.3 eV, which corresponds to 100% of C sp^3^.

Raman spectroscopy in the 1000–1700 cm^−1^ range (Figure 3d) revealed that the samples (2T-9010 and 2T-8020) were able to recover completely thanks to thermal annealing from induced strain after laser processing [23]. This is due to the low accumulated fluence absorbed in the second texturing step of the fabrication process, which has a greater influence on surface restructuring thanks to the incubation effect; in fact, samples with a higher energy dose retained a good percentage of strained covalent bonds. We extended the Raman measurement to wavenumbers up to 1700 cm^−1^ in order to evaluate the possible presence of graphite formation, but there was no evidence of graphite clusters on the surface after the sample cleaning; therefore, graphite has to be excluded as a main actor of the present conductivity.

### 3.3. Cryogenic Measurements

In order to assess the electrical transport properties of the BD samples we evaluated four different resistances, calculated as the ratio between the voltage difference measured at the two opposite contacts of the injecting ones and the injected current amplitude, according to the method illustrated by L.J. Van der Pauw [28]. All the four resistance values acquired during the (i)–(iv) phases were measured as a function of the sample temperature. Figure 3 and Figure 4 show the experimental results for the 2T-9010 and 2T-8020 sample, respectively. Each plot reports data acquired for the R_hk,ij_ resistance, where h–k indicate the current-injection contacts, whereas i, j are the two contacts located in the opposite side used for the voltage measurement. Resistance measurements were performed on both the cooling and heating periods. Blue diamonds refer to the data acquired cooling the samples from room temperature (RT) to 30 K. Conversely, red diamonds represent the results obtained during the following sample-heating from 30 K to RT. In Table 1, the resistances measured at 300 K and 30 K are also summarized.

At room temperature, the 2T-8020 sample is slightly more conductive in comparison to the 2T-9010. This result is in good agreement to the measurements performed at a high temperature on the same specimens [23]. As it is possible to observe in Figure 4 and Figure 5 and Table 1, the resistance value strongly decreases along with the measurement stage in both samples of about a decade at RT and of almost a factor 20 at the minimum temperature of 30 K. Furthermore, for both samples, the most conductive stage is the second at RT and the last one at 30 K. As clearly illustrated by results reported in Figure 4 and Figure 5, the monotonic decrease of the resistance with temperature (as expected for a charge transport mechanism of a semiconductor) is not observed for all the measured resistances. In particular, R_34,12_ of 2T-9010 (Figure 4c) and R_23,41_ of 2T-8020 (Figure 5b) display a clear transition from a negative-temperature-coefficient (NTC) to a positive-temperature-coefficient (PTC) at temperatures lower than 100 K. 

Around the same temperatures, the sample 2T-8020 shows a weak NTC-to-PTC transition also in the R_34,12_ resistance. The sample 2T-9010 behaves like PTC immediately cooling down from RT in the R_23,41_ resistance. These transitions are more or less exactly repeated in the successive heating measurement phases, thus clearly indicating that such behavior is strictly related to the DB surface hopping transport.

The last enthralling observation is that, except for the weakest transition of sample 2T-8020 at the third measurement stage, the NTC → PTC transition is always related to a loss of the conductance (i.e., a resistance greater than 120 MΩ, the maximum value detected by the Keithley ohmmeter coupled to the multiplexing board). In the following discussion, we will define the decrease of the conductance values of more than three decades as an insulator transition (as already indicated in Figure 4 and Figure 5) mimicking the Mott definition for transition metal oxides like NiO [29]. If we analyze the occurrences of the insulator transitions, we can observe, as evident in Table 2, that it occurs in a short range of conductivity values across all the temperatures: 125–168 kΩ for 2T-9010 and 190 kΩ for 2T-8020, which correspond to the maximum values.

## 4. Discussion

The experimental results reported in the previous section and summarized in Figure 3 and Figure 4 emphasize the difficulty of deriving the sheet-resistance of the material by a simple application of the Van der Pauw method. However, the data recorded at each temperature and acquired by the four-phase current-voltage measurements allowed us to highlight the peculiar behavior of the samples. R_12,34_ acquired on phase (i) displays a unique monotonic decrease in the 30–300 K range, for both the two samples, at each temperature, and for both the cooling and heating cycles. In order to get more insight into the conduction mechanisms of the nanotextured surface, Figure 6 reports the Arrhenius plot of conductance, previously illustrated as resistance in Figure 3a and Figure 4a. The experimental results clearly show the presence of two activation regimes with a slope change around 50 K. In order to evaluate the activation energies in the cryogenic temperature range for the two samples, we considered two thermally activated populations of conducting charges. The conductance of the samples can be expressed as: (2)$$G=G_{PreF_{1}}\cdot e^{-\frac{E_{a1}}{k_{B}\cdot T}}+G_{PreF_{2}}\cdot e^{-\frac{E_{a2}}{k_{B}\cdot T}}=G_{PreF_{1}}\cdot e^{-\frac{E_{a1}}{k_{B}\cdot T}}*\left(1+r_{G}\cdot e^{-\frac{∆E_{a}}{k_{B}\cdot T}}\right)$$

We can consider a first low-temperature region up to 50 K (i.e., 1000/*T* < 20 K^−1^) where data of Figure 5 are dominated by the first term of Equation (2). An activation energy *E*_*a*1_ of a few meV is estimated. The best fit procedure has then been applied to the entire set of data to evaluate the last two parameters *E*_*a*2_ and *G*_*PreF*_2__. The calculated values are reported in Table 3. 

Both samples show an activation energy of a few meV in the 30–50 K range and a few tens of meV at higher temperatures. The pre-exponential factor *G*_*PreF*_1__ around 80 nS is evaluated in both the 2T-9010 and 2T-8020 samples. Conversely, in the second activation energy temperature range, the pre-exponential factor related to the density of defective states in the 2T-8020 sample is about 50% higher than the one found for the 2T-9010, making this sample the more conductive at RT. The low value of the estimated activation energy for the electrical conduction is in agreement with previous reports of polycrystalline CVD diamond films [30,31], assuming Mott Variable Range Hopping (VRH) mechanism charge transport with activation energies ranging from 15.6 meV to 228 meV. This range of values depended on the film-growth procedure and then on the density of crystalline defects in the diamond thin film. Therefore, we can definitively state that the activation energy literature values reported in literature are in good agreement with those found for the samples investigated in this work (32–37 meV), underlining a similarity between the electrical transport in the polycrystalline CVD-diamond columnar structure and the BD LIPSS in the *E*_*a*2_ region.

As stated in Section 2, the reduction of the nanostructuring alignment is more evident in the 2T-8020 sample, where a higher amount of laser radiation dose was released during the second Y-scan (see Figure 2b); therefore, 2T-8020 is expected to be more defected if compared to the 2T-9010. In this case, the percolation path of the hopping transport is dominated by a high deviation from the mean required energy distance between adjacent impurity centers, resulting in a higher value of the activation energy. In addition, the density of states within the bandgap in this sample is expected to increase along with the amount of absorbed laser-energy due to the incubation effect. As illustrated by data reported in Table 3, compared to the 2T-9010, the 2T-8020 sample shows both a higher activation energy and a higher density of defect states in the *E*_*a*2_ region. 

In the lowest temperature range (30–50 K), even lower activation energy values are found (4–6 meV). Such a result is ascribed to the transport-activation by ultra-shallow states in the diamond bandgap due to structural defects created by the fs-laser treatment. Such low values of activation energy were found in n-doped ultra-nanocrystalline diamond (UNCD) [32], and they were ascribed to a concentration of nitrogen atoms in the grain boundaries, leaving dangling bonds associated with available states near the Fermi level. According to the similarity of values found for the conductance and the activation energy for both the two samples, it is possible to state that fs-laser processing introduces similar ultra-shallow centers in the bandgap and a nitrogen contamination induced by the extremely high-temperature developed during the laser treatment cannot be ruled out. 

In addition to shallow defects, laser treatment also introduces deeper defects into the diamond bandgap. Acting as recombination centers, deeper defects justify the differences of the R_hk,ij_ values found at each temperature on stages from 1 to 4 (see Figure 4 and Figure 5). This results in an enhancement of the charge mobility through the contacts and then a decrease of the measured resistance according to the free traps’ energy distribution:(3)μ=μ01+exp[EtkT]
where *µ* is the effective mobility and *µ*_0_ is the ideal mobility with all the traps filled. Therefore, at RT when a larger current is flowing (the ratio I(RT)/I(30K) is 44 for 2T-8020 and 21.2 for 2T-9010), we are able to fill more traps in one single measurement and a higher conductivity is observed at the second stage, while at 30 K, we need the completion of all the four stages to reach the most favorable condition. We can also observe that for 2T-9010, the maximum conductivity at 30 K is practically obtained already at the third stage: 2.65 MΩ vs. 2.6 MΩ at the fourth. In fact, at 30 K, 2T-9010 is able to transport the double of the current of 2T-8020. Since the Keithley 7700 multiplexer module opens the measurement channel through electromechanical relays with an actuation time of about 3 ms and a measurement window of about 200 ms, we may estimate an emission time for the trapped charge at least of one second around RT, since they have to keep it trapped for all four of the stages of the measurement cycle, which lasts about 1s. On the other hand, when the software system waits to start the acquisition of the following measurement cycle at the new desired temperature, the conductivity enhancement seems to vanish, including in the consecutive tests at room temperature before any temperature change. Therefore, we can estimate that charges are not lasting more than a few minutes in the trap centers at RT, since this was the fastest interval time between two acquisition cycles.

Strictly related to the electron trapping is the possibility to have exchange interaction between electrons confined on the black diamond’s surface, thus generating spin-related transport effects explaining the evident NTC → PTC transitions in the intermediate stages of the single measurement. It has recently been demonstrated by Esquinazi’s group [33] that nitrogen doping induces unconventional magnetization below 25 K in diamond. Previous research works [28] demonstrated that ferromagnetism is observed even at RT in nitrogen-doped nano-crystalline diamond films [34]. Zhang et al. [35] also observed the electronic entanglement between superconducting and ferromagnetic states in hydrogenated boron-doped nano-diamond films, with a Curie temperature higher than 400 K. Similarly to what happens in manganese oxide compounds (MnO_3_) [36,37], where the hopping transport is ruled by a double exchange mechanism between lattice atoms with different orbital occupancies and the transport is favored by the geometrical ordering of the electron spin, in the BD surface at stages 2 or 3, where it has been supposed that electrons are trapped on the surface, a NTC → PTC transition is expected after the temperature is lowered under the Curie one since electrons will move by correlated hopping. According to the values extrapolated in Table A1, the Curie temperature is higher at the second stage of the measurements, indicating a negative effect of the amount of trapped charge on its value or presence (at stage 4 in both samples, the NTC → PTC transition is not even present). This again corresponds to the behavior of ferromagnetic oxides, which have a maximum Curie temperature according to stoichiometry ratio/charge doping variation [38].

Regarding the metal–insulator transition successive to the NTC → PTC transition, since as illustrated in Table 3, it happens in a short range of conductivity values, we correlate it to a critical charge density over the BD surface [39]. At this level of charge density, the electrons on the surface may find an antiferromagnetic (AF) ordering of the spin energetically convenient between the adjacent hopping center with insulating behavior. In fact, at that critical charge density, the exchange interaction begins to dominate the Hubbard–Hamiltonian and electron distributions over the lattice sites uniformly, with the condition that every site has to carry exactly one electron to avoid a potential energy increase. This electron arrangement evidently suppresses the hopping transport. Further proof of the critical charge density assumption is the case of R_34,12_ measurement of the 2T-8020 sample, the only NTC → PTC region where the metal–insulator transition does not occur. In this case, in the range of 80–100 K, the resistance reduces only from 1.7 MΩ to 1.3 MΩ, which corresponds to a current density about one decade lower on behalf of the other NTC → PTC regions where the metal–insulator transition takes place (see Table 3). Reducing the temperature further, the BD loses the insulating phase and it again becomes semiconducting, with a resistance restarting from values close to the maximum ones (Table 3, last column). Such behavior is expected, since at low temperature, the current injection during the first stage is reduced (even if with very low activation energies as aforesaid) and therefore, the critical charge density is not reached anymore.

## 5. Conclusions

We performed electrical conductivity measurements in the temperature range from 30 K to about 300 K for two 2T-BD samples obtained by thermal-grade polycrystalline diamond films. LIPSSs were produced on the surface along two perpendicular directions with a similar modality of a laser-energy release. The analysis of the activation energy highlighted a similar behavior for the samples with a further decrease in the already low value measured in the range 300–500 K: from about 30 meV to 5 meV, thus attributing a considerable surface conductivity to the BD films even at the cryogenic temperatures.

The cycled investigation of the samples’ transport properties evidenced a change in the conductivity over the same surface according to subsequent measurement stages, thus paving the way to the possibility of tuning diamond surface transport properties even at RT. This optimization may be especially important for sensor applications, where diamond surfaces are widely used for DNA biosensing thanks to its exceptional biocompatibility.

Furthermore, in the cryogenic temperature range, we observed typical mechanisms of the NTC→PTC transitions, suggesting the presence of spin-related electron exchange interactions. Linked to the previous phenomenon, we also demonstrated the presence of sudden insulator transitions. Such behavior opens the path to look for better sensing capabilities even at cryogenic temperatures, which is also where biomaterials like antibodies or enzymes are usually stored. A further investigation should look for the same phenomena in distinctive diamond surface nanotexturing with an enhanced aspect ratio [14].

## Figures and Tables

**Figure 1 nanomaterials-12-02253-f001:**
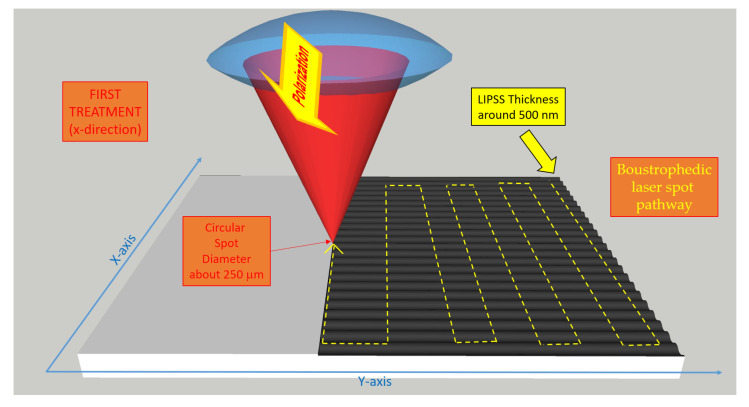
A 3D-view of the two separate boustrophedic fs-laser treatments of the diamond surface. The LIPSS nano-waves’ direction is orthogonal to the light polarization and their thickness is about half a micron. In the lower figure, the larger optical absorbance of the diamond surface after the second treatment is evidenced due to the increased defects’ densities [22].

**Figure 2 nanomaterials-12-02253-f002:**
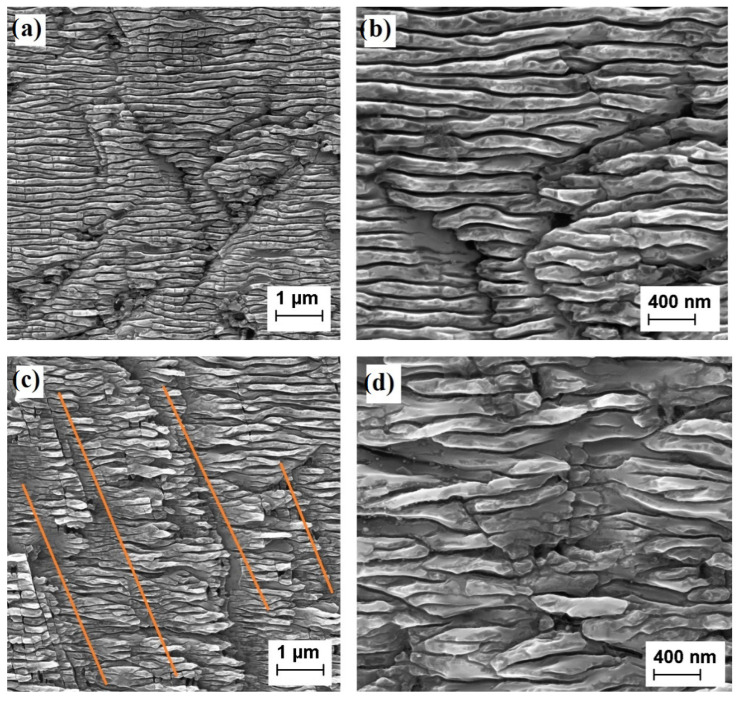
Top view of the central area of samples 2T-9010 (panels (**a**,**b**)) and 2T-8020 (panels (**c**,**d**)).

**Figure 3 nanomaterials-12-02253-f003:**
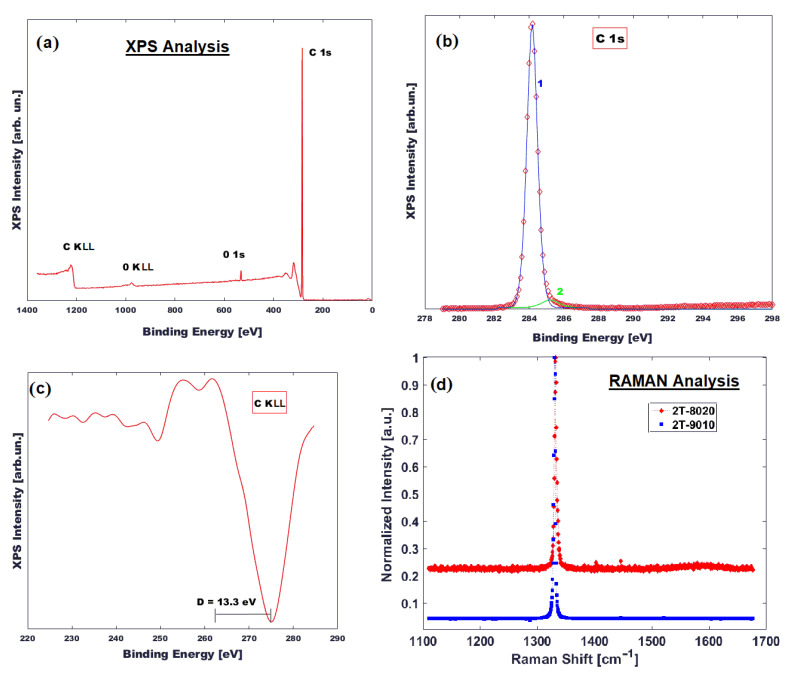
Surface characterization of BD samples: XPS (panels (**a**–**c**) and Raman (panel (**d**))).

**Figure 4 nanomaterials-12-02253-f004:**
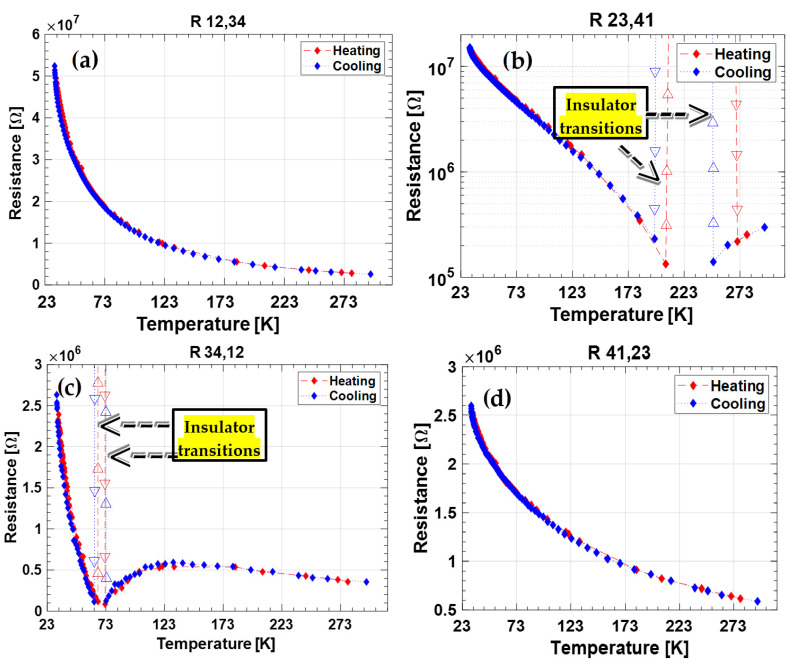
Measured resistance of sample 2T-9010 during the characterization performed following the Van der Pauw method. (**a**) Current injected at contacts 1–2; (**b**) current injected at contacts 2–3; (**c**) current injected at contacts 3–4; (**d**) current injected at contacts 4–1. The figure relative to the second cycle of both samples is represented on logarithmic y-axes in order to have eye-evidence over the NTC → PTC transition.

**Figure 5 nanomaterials-12-02253-f005:**
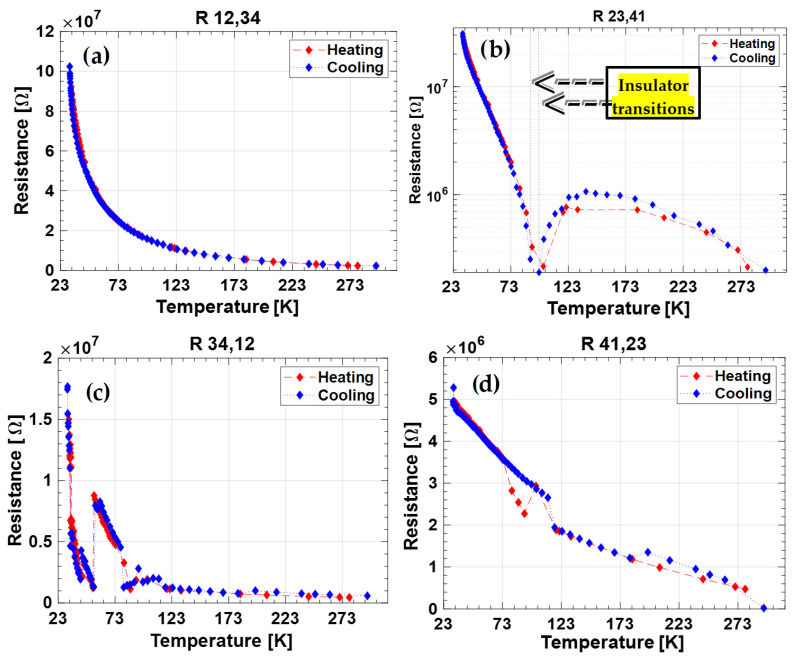
Resistance of sample 2T-8020 at different contacts: (**a**) current injected at contacts 1–2; (**b**) current injected at contacts 2–3; (**c**) current injected at contacts 3–4; (**d**) current injected at contacts 4–1. The figure relative to the second cycle is represented on logarithmic y-axes in order to have eye-evidence over the NTC → PTC transition.

**Figure 6 nanomaterials-12-02253-f006:**
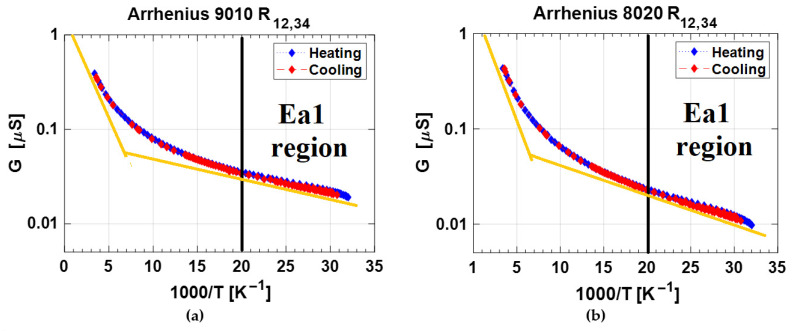
Arrhenius plot of conductance of (**a**) R_12,34_ of sample 2T-9010, and (**b**) R_12,34_ of sample 2T-8020.

**Table 1 nanomaterials-12-02253-t001:** Resistance values measured at the highest (300 K) and the lowest (30 K) temperature during the four stages of current injection.

Stage	Injection Points	8020 @ 30 K	8020 @ RT	9010 @ 30 K	9010 @ RT
1	I → 1–2	102 MΩ	2.3 MΩ	53 MΩ	2.5 MΩ
2	I → 2–3	31 MΩ	210 kΩ^1^	15.2 MΩ	300 kΩ
3	I → 3–4	18 MΩ	450 kΩ^1^	2.65 MΩ	350 kΩ
4	I → 4–1	5.3 MΩ	480 kΩ^1^	2.6 MΩ	580 kΩ

**Table 2 nanomaterials-12-02253-t002:** Temperature and resistance values at the moment of metal–insulator transition of the samples and of the insulator–semiconductor back transition.

MEASUREMENT	Stage	Temperature (K)	Resistance (kΩ)
2T-9010 Cooling	3	74 → 63	125–115
2T-9010 Heating	3	64 → 81	168–241
2T-9010 Cooling	2	248 → 196	140–232
2T-9010 Heating	2	206 → 276	134–219
2T-8020 Cooling	2	98 → 90	190–252

**Table 3 nanomaterials-12-02253-t003:** Calculated values for *E*_*a*1_, *E*_*a*2_, *G*_*PreF*_1__, and *G*_*PreF*_2__ following the best fit of data of Figure 5 according to Equation (2).

MEASUREMENT	30–50 K	50K–RT
2T-9010	*E*_*a*1_ = 4 meV; *G*_*PreF*_1__ = 85.1 nS	*E*_*a*2_ = 31.4 meV; *G*_*PreF*_2__ = 0.94 µS
2T-8020	*E*_*a*1_ = 5.6 meV; *G*_*PreF*_1__ = 80.9 nS	*E*_*a*2_ = 35.3 meV; *G*_*PreF*_2__ = 1.4 µS

## Data Availability

The data presented in this study are available on request from the corresponding author. The data are not publicly available due to university policies.

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
