# Peer review of "Charge Transport Mechanisms of Black Diamond at Cryogenic Temperatures"

_nanomaterials, 2022, doi:10.3390/nano12132253_

Round 1
Reviewer 1 Report
My overall impressions of the general quality and the scientific soundness of the paper are positive. The authors provided here a detailed characterization of the investigated materials. The material is interesting and important from an application point of view. In my opinion, the presented results are of good quality and well-described. Therefore, the paper deserves to be considered for publication. Also, all tables, figures, and formulas are provided with good quality. The language and the style are acceptable. The article is carefully written. I have found only a few errors/mistakes. Before the acceptance, the author should follow the comments below:
- In line 76 missing notation of sheet resistance. In lines 102, 114, and 247, there is “both the two samples”. Both samples should be enough.
- What was the size and shape of the laser spot on the sample? Also, how was the sample positioned perpendicularly to the laser? In such an experiment, it is crucial to keep the system's geometry all the time. Any deviation can change the morphology of the sample.
- How deep does the laser penetrate the sample? The question here arises: What are the modified surface's thickness and the substrate's influence on the resistivity measurements? The samples are in bulk form, but the surface was modified, and I suppose this is a quite thin layer. If the layer is very thin, it is not easy to measure its properties without considering what is below.
- Another thing concerns me. The samples exhibit different behavior and properties. In my opinion, that means the surface properties strongly depend on the preparation. It could be interesting to check the repeatability, which is very important in front of a view of the application in any sensing measurements.
According to the comments above, I recommend a major revision of the paper.
Reviewer 2 Report
This manuscript investigates the electronic transport mechanism of fs-laser nanotextured diamond under cryogenic temperatures. The activation energy of a few tens of meV in the highest temperature interval was found. However, there is a minor point that needs to be clarified. The authors should address it to be qualified to publish in the Nanomaterials
- The authors immersed the laser-ablated samples in a strongly oxidizing solution to remove debris and non-diamond carbon phases. The morphologies before and after cleaning should be presented to show the changes in micro-nanostructures. Moreover, the chemical compositions (by EDS) and the chemical bondings/structures (XRD, FTIR, XPS) on the surfaces before and after immersing in the solution should be investigated to ensure that the mechanism is coming from only LIPSS formation or both effects from new structure formation and chemical modification.
Reviewer 3 Report
The manuscript by A. Orsini et al. reports on temperature dependent transport measurements of so-called black diamond,
i.e. CVD diamond films that are treated by fs laser pulses in order to introduce defects. While I was rather curious about the topic and the announced results, going through the manuscript I more and more got disappointed and finally have to conclude that the paper cannot be published.
Some of the reasons are listed in the following, however, many other points could be added:
1. The preparation of the samples is unclear: How is the nanotexturing done? In particular what is meant by structuring in x-y direction? Is the sample simple rotated by 90 degree and the one-dimensional LIPSS structure done again?
2. The results of the resistivity measurements are inconsistent. For a rectangular sample with for contacts in the four corners, I expect that the results are similar when current is injected in 1-2 and voltage is probed at 3-4 compared to measurements with current injected in 3-4 and voltage measured at 1-2. Here the results different by orders of magnitude. The same problem occurs for 2-3 and 4-1. The same occurs for both samples.
3. The temperature dependent results should be similar when looking at configuration 1 compared to configuration 3. I am afraid I missed an important point.
4. There are temperatures where the data exhibit some divergency in the sense that the resistance vanishes for some time. The authors call this a metal-insulator transition, without any reasons given.
5. The data in Fig. 4 exhibit some issues, jumps and kinks which are not discussed at all. What is going on here?
6. The Arrhenius plots in Fig. 5 clearly show a smooth behavior of the data. The fit by two exponential curves is not convincing. As a matter of fact, variable range hopping does not follow a simple activated behavior but can better be described by a power law, where the exponent depend on the relevant dimension, which might be one, two or three here.
7. The rest of discussion becomes even more unclear and I had to stop my efforts understanding it.
In conclusion, the manuscript is unsuitable for publication.
Round 2
Reviewer 1 Report
After carefully reading the authors' responses, I would say I am satisfied. I think the work was improved enough to warrant the publication in Nanomaterials.
Reviewer 3 Report
no comment